# The high price of equity in pulse oximetry: A cost evaluation and need for interim solutions

**Katelyn Dempsey**[1], **Joao Matos**[1], **Timothy McMahon**[1,2], **Mary Lindsay**[3], **James E. Tcheng**[4], **An-Kwok Ian Wong**[1,2,5]\*

1 Department of Medicine, Division of Pulmonary, Allergy, and Critical Care Medicine, Duke University, Durham, North Carolina, United States of America, 2 Durham Veterans Affairs Hospital, Durham, North Carolina, United States of America, 3 Duke University Hospital, Durham, North Carolina, United States of America, 4 Department of Medicine, Division of Cardiology, Duke University, Durham, North Carolina, United States of America, 5 Department of Biostatistics and Bioinformatics, Division of Translational Biomedical Informatics, Duke University, Durham, North Carolina, United States of America

\* a.ian.wong@duke.edu, med@aiwong.com

**Data Availability Statement:** Equipment list is included in S1 Table above 1% prevalence for Duke University Health System as per Duke Health IRB approval under Pro00113724.

## Abstract

Disparities in pulse oximetry accuracy, disproportionately affecting patients of color, have been associated with serious clinical outcomes. Although many have called for pulse oximetry hardware replacement, the cost associated with this replacement is not known. The objective of this study was to estimate the cost of replacing all current pulse oximetry hardware throughout a hospital system via a single-center survey in 2023 at an academic medical center (Duke University) with three hospitals. The main outcome was the cost of total hardware replacement as identified by current day prices for hardware. New and used prices for 3,542/4,136 (85.6%) across three hospitals for pulse oximetry devices were found. The average cost to replace current pulse oximetry hardware is $6,834.61 per bed. Replacement and integration costs are estimated at $14.2–17.4 million for the entire medical system. Extrapolating these costs to 5,564 hospitals in the United States results in an estimated cost of $8.72 billion. "Simply replacing" current pulse oximetry hardware to address disparities may not be simple, cheap, or timely. Solutions for addressing pulse oximetry accuracy disparities leveraging current technology may be necessary, and might also be better.

**Trial Registration**: Pro00113724, exempt.

## Author summary

Recent evidence indicates patients of color are negatively influenced by discrepancies in pulse oximetry, thus leading to critical health outcomes. Due to these findings, we provide a rough estimate of the cost and complexity of replacing current pulse oximetry technology to improve disparities, both at a single institution and across the US. We utilized an observational study of pulse oximetry devices in an academic medical system with three hospitals, new and used prices were found for 3,542/4,136 devices (85.6%), with a total replacement and integration cost of $14.2–17.4 million and some life cycles extending beyond 18 years. When extrapolated to 5,564 hospitals in the United States, estimated

**Funding:** AIW is supported by the Duke CTSI by the National Center for Advancing Translational Sciences of the NIH under UL1TR002553 and the National Institute on Minority Health and Health Disparities REACH Equity Award under 5U54MD012530. TM is supported by NHLBI R01 HL161071. The funders had no role in study design, data collection and analysis, decision to publish, or preparation of the manuscript.

**Competing interests:** AIW holds equity and management roles in Ataia Medical. All other authors have declared that no competing interests exist.

replacement costs are $3.57-$11.5 billion. In general, the monetary and labor cost of pulse oximetry hardware replacement is substantial, and solutions utilizing current pulse oximetry technology are essential to delivering equitable care to all patients.

## Background

Pulse oximetry is a critical tool in modern medicine that provides a continuous and real-time measurement of arterial oxygenation, a critical marker of respiratory function [1]. However, in patients with dark skin pigmentation–especially Black patients–pulse oximetry accuracy disparities have been demonstrated, as evidenced by increased hidden or occult hypoxemia, which was first identified in December 2020 [2,3]. These inaccuracies are associated with increased mortality, organ dysfunction, decreased oxygen therapy, and delayed COVID recognition [3–5].

In February 2021, the Food and Drug Administration (FDA) issued a safety communication calling on patients and health care providers to be aware that pulse oximeters have critical limitations [6]. Many have called for simply replacing all pulse oximetry devices with new technology; as a consequence, the FDA is engaging with multiple stakeholders to reassess existing pulse oximetry clearance guidelines [7,8]. There are early possibilities, such as green light oximetry, cameras, and improved algorithms [9–11], that are supported by alternative physical principles to estimate arterial oxygen saturation in a way that is not sensitive to skin pigmentation. These novel devices will likely require new equipment for both pulse oximetry probes and monitors. However, "simple replacement" of existing devices may not be simple, cheap, nor timely [12]. Flawed pulse oximeters are still ubiquitously used at present, including home, ambulatory, and hospital settings [13]. Moreover, pulse oximeters are often integrated into hospital-wide complex electronic health records (EHR) systems; current medical device ecosystems often require vendor-specific interfaces that hamper interoperability and integration of new devices [14].

Pulse oximetry equipment falls into three categories: multi-parameter models (e.g, Philips MX800, GE Dash 4000, Philips X3) are complex units that display multiple vital signs, predominantly used in ICUs and operating rooms; Pulse Oximetry Modules/Monitors (e.g, Masimo Rad-7) are specialized for pulse oximetry and can be integrated into existing systems without replacing whole units; and Vital Signs Monitors (e.g, Philips SureSigns VS4) are primarily used in outpatient and general inpatient settings, collecting various vital signs (see Table 1 for examples). While monitors may be integrated with EHR systems, automatically performing data flows, standalone devices require manual data entry. Among these, module-based devices are the easiest to replace.

The objective of this study is to describe the financial and logistical burden in replacing current pulse oximeters at one health system with three hospitals to provide a cost estimation of replacing all pulse oximeters across the United States [2,9].

## Methods

### Equipment data

The Duke University Health System provided extensive equipment inventories, categorized by individual locations. New and used equipment prices were sought online, primarily utilizing Google and comprehensive sites like Bimedis.com. Price data was retrieved in June 2023. All prices were calculated in United States dollars ($). Data on hospital size was retrieved from the

**Table 1. Pulse Oximetry equipment categories and examples.** Example device images are included in S2 Fig.

| Device | Description | Examples | Ease of replacement |
|---|---|---|---|
| Multi-Parameter Models | Complex units that display multiple vital signs | Philips MX800 | • Often wall-mounted and has a rack to accept multiple modules, including arterial pressures, pulmonary artery pressure, and other modules (e.g., the bispectral index [BIS], etc). <br> • Requires the X3 (or similar module) to function. |
| | | Phillips X3 | • This device is portable and utilizes a portrait/landscape display <br> • The X3 can function independently as a temporary transport monitor as long as its battery capacity allows. |
| | | GE Dash 3000 | • This device is portable and designed for easy handling. <br> • However, since the pulse oximetry is integrated, the entire device must be replaced. |
| Pulse Oximetry Modules and Monitors | Specialized for pulse oximetry, independent from other modules, can be integrated into existing systems without replacing whole units | Masimo Radical-7 | • The device is detachable/portable for patient transport and rotates depending on orientation <br> • Measures arterial oxygen saturation (SpO2), pulse rate, and perfusion index (PI). Other measurements include hemoglobin (SpHb), carboxyhemoglobin (SpCO), methemoglobin (SpMet), total oxygen content (SpOC), pleth variability index (PVI), and acoustic respiration rate (RRa) <br> • High ambient light sources can interfere with the performance of the sensors [15]. |
| | | GE Patient Data Module | • Requires a display to function (e.g., GE Criticare) <br> • Monitors temperature, invasive pressures (e.g. ART, FEM, PA, etc.), non-invasive pressures (e.g. heart rate, systolic and diastolic pressure, etc.), and pulse oximetry |
| Vital Signs Monitors | Collect various vital signs (often temperature, blood pressure, heart rate, pulse oximetry) | Philips EarlyVue VS30 | • Often on a stand, wheeled from room to room <br> • Staff tend to type the numbers from the monitor into the EHR, however these devices can be tailored to a particular clinical setting. <br> ○ Note that some models may now support device integration [16]. <br> • Touch screen display monitoring noninvasive blood pressure, SpO$_2$, pulse, CO$_2$ and respiration rate (from site commented) |

American Hospital Directory. To reconcile new and used prices, the lowest ratio of new or Manufacturer's Suggested Retail Prices (MSRP) to used prices was conservatively estimated from devices with both new and used prices. No discount rate was assumed given the lack of transparency to discount rates.

The Duke University Health System IRB approved Protocol Pro00113724 as exempt.

## Device integration estimates

The integration process involves multiple steps, including server specifications, firewalls, and software revisions. Estimates on integration time required by Duke Performance Services leadership, which oversees device integration, recommended 250–500 person-hours per device model, at a rate of $200/hour. Costs were fully applied to multi-parameter monitors, 50% to pulse oximetry modules/monitors, and not applied to vital signs monitors.

## Extrapolation to United States National Hospital data

Hospital data from 2015 was obtained from the Health, United States data finder [17]. Total (i.e. the complete collection of pulse oximetry-related equipment utilized in a hospital, including the devices themselves and any accompanying apparatus) replacement costs were modeled in two ways: based on estimated costs per bed by at Duke University Hospital, Duke Raleigh Hospital, and Duke Regional Hospital and based on used price percentages of MSRP.

We utilized 2015 national hospital data from the Health, United States data finder. Estimates for total replacement costs were based on two models: one using cost per bed by types of hospitals and the other using used price percentages of MSRP.

## Results

### Cost estimates for a single academic health system

Duke University Health System (DUHS) is comprised of three main facilities. (S1 Fig) Duke University Hospital (DUH) is a large, quaternary, academic center with 1,048 beds, including 332 designated for specialized care (i.e., defined as a hospital bed in which special medical/surgical services, beyond general medical/surgical care and including intensive care or coronary care, are provided [18]). Duke Raleigh Hospital (DRaH) is a medium-sized facility with 186 beds, of which 28 are specialized. Duke Regional Hospital (DRH) has 373 beds, with 22 allocated for specialized care. Across these hospitals, 140 distinct models of pulse oximetry equipment are in use (as seen in S2 Table). Of all 8,460 devices at Duke University Health System (DUHS), 5,678 (67.1%) devices are used in hospitals. 1,542 of these devices are multiparameter monitors that have their pulse oximetry hardware in external monitors or modules (e.g., MX450, MX800 as the multiparameter monitor, relying on the Phillips X2 or X3 multiparameter monitor/module for pulse oximetry hardware. See Table 1 for examples).

Two key devices, Philips SureSigns VS4 and MX700, have both new and used pricing available. The ratio of new prices to used prices for these devices ranges from 513% to 133% (see S1 Table for prices). Consequently, the new Manufacturer's Suggested Retail Price (MSRP) was conservatively approximated as 133% of the used prices, with an adjustment range of 110% to 200%. Of the 4136 devices currently in hospital use, 85.6% had either new or used price data. The estimated replacement cost for these devices, not including integration costs, is approximately $11 million. This estimate encompasses 85.6% of devices currently in use, with per-bed costs ranging from $3,980 to $12,770.

Additional costs for integrating these devices are projected to be between $3.25 and $6.5 million, leading to a total projected cost of $28.5 to $31.8 million for the hospitals. When extending the scope to include clinics affiliated with Duke University Health System, the estimated replacement cost for all devices rises to $35.9 million. With projected integration costs of an additional $4.4 to $8.8 million, the overall expenditure is estimated to be between $40.3 and $44.7 million.

### Extrapolation to national costs

National United States hospital data from the CDC 2015 data reveals 5,564 hospitals, with an average of 161.4 beds/hospital. Estimates looking at different base hospital rates, assuming 75%, are in S3 Table. At an estimated cost from $3,975–12,770/bed for total replacement, these estimates suggest that total replacement will cost $9.7–20.1 billion.

Using an average of all rates across the health system, the new MSRP can be estimated from used costs. Based on an estimated cost of $6,835/bed ($3,975–12,772/bed), the estimated total replacement cost is $14.1 billion ($13.7–22.1 billion) (S4 Table).

## Discussion

The racial disparities in pulse oximetry accuracy are not just statistically alarming but also clinically urgent. Such disparities contribute to increased morbidity and mortality among patients of color, a pressing issue that cannot be relegated to the lengthy timelines associated with natural equipment replacement cycles [19]. Our analysis reveals that full equipment overhaul is neither cost-effective nor timely, thereby mandating prioritized attention at federal and industry levels for alternative solutions.

There is an immediate need for targeted funding and policies that can drive rapid innovation in this space. Federal agencies must take the lead in establishing frameworks that

incentivize the development of more accurate, yet economically viable, pulse oximetry technologies. Similarly, industry stakeholders should prioritize research and development investments towards creating lower-cost solutions that can be deployed quickly.

### Interim solutions

Informatics-based approaches are especially promising in this context. By leveraging existing hardware, existing EHR pipelines, and data analytics, heath systems could implement more accurate and less invasive solutions sooner, without having to wait for the end-of-life of current devices to replace existing hardware. Specifically, we envision the use of machine learning (ML) models that, based on a patient's trajectory since admission (patient characteristics, vital signs, laboratory test values, and specific treatment information), could predict the risk of hidden hypoxemia or estimate the actual arterial oxygen saturations [20]. Such models would act as an early warning system to address the shortcomings that existing pulse oximetry devices present.

These strategies not only avoid the massive financial burden of full-scale equipment replacement—estimated up to $22.1 billion nationally—but also present an opportunity for quicker implementation. However, such approaches would benefit substantially from federal and industry support, be it in the form of research grants, tax incentives, or regulatory fast-tracking.

### Limitations

The study has limitations, such as incomplete price data and unaccounted volume discounts, which may underestimate the actual replacement costs. Also, device integration costs may be underestimated, particularly for vital signs monitors, considering inflation, other interfacing hardware, and staff retraining costs that may be applicable. Finally, we note that 1,542 (27.2%) multiparameter monitors (e.g., Philips MX400, MX450, etc.) did not need to be replaced due to pulse oximetry hardware in an external module (e.g., Phillips IntelliVue X2, X3); if another health system had monitors that had integrated pulse oximetry, it would add another $14.2 million in equipment costs, leading to an average cost increase in $8,870 per bed (to a total of $25.2 million across the system and an average per-bed cost of $15,704).

### Conclusion

The racial disparities in pulse oximetry accuracy have immediate and dire clinical ramifications. It is unlikely that there is a pure software upgrade solution or a backwards-compatible probe as a "drop-in" replacement. Consequently, a simplistic notion of device replacement is neither affordable nor timely, demanding an estimated $9.7–22.1 billion and several years for nationwide implementation. Given these challenges, it is essential to prioritize federal and industry interventions that can innovate lower-cost and quicker solutions, such as informatics-based approaches. Policymakers, healthcare providers, and researchers must urgently collaborate to navigate this complex problem efficiently and equitably.

### Supporting information

**S1 Fig. Flow diagram.** Flow diagram for all devices at Duke University Health System, focused on in-hospital devices for acute care. Note that 1,542 multiparameter monitors were excluded as they would not need to be replaced, as their pulse oximetry was measured in modules. Furthermore, note that for DUH, DRaH, and DRH, the sum of new (1436) and used equipment with prices (2458) is higher by 336 (3,542 with prices), as 336 devices had both new and used

prices.
(TIFF)

**S2 Fig. Example images of devices referenced in Table 1.** Example of devices referenced in Table 1. a. Multiparameter module: Philips MX800. b. Multiparameter module: Philips X3. c. Multiparameter module: GE Dash 3000. d. Pulse oximetry modules and monitors: Masimo Radical-7. e. Pulse oximetry modules and monitors: GE Patient Data Module. f. Vital signs monitors: Philips EarlyVue VS30.
(TIFF)

**S1 Table. List of price estimates for devices.**
(DOCX)

**S2 Table. Average replacement cost per bed.**
(DOCX)

**S3 Table. Device integration costs.**
(DOCX)

**S4 Table. National estimates in fleet replacement cost, using different used/new price estimates, assuming average prices across health system.**
(DOCX)

## Author Contributions

**Conceptualization:** An-Kwok Ian Wong.

**Data curation:** An-Kwok Ian Wong.

**Formal analysis:** An-Kwok Ian Wong.

**Funding acquisition:** An-Kwok Ian Wong.

**Investigation:** Katelyn Dempsey, Mary Lindsay, James E. Tcheng, An-Kwok Ian Wong.

**Methodology:** Katelyn Dempsey, An-Kwok Ian Wong.

**Project administration:** Katelyn Dempsey, An-Kwok Ian Wong.

**Resources:** Mary Lindsay, An-Kwok Ian Wong.

**Software:** An-Kwok Ian Wong.

**Supervision:** An-Kwok Ian Wong.

**Validation:** Katelyn Dempsey, Joao Matos, An-Kwok Ian Wong.

**Visualization:** Joao Matos, An-Kwok Ian Wong.

**Writing – original draft:** Katelyn Dempsey, An-Kwok Ian Wong.

**Writing – review & editing:** Katelyn Dempsey, Joao Matos, Timothy McMahon, Mary Lindsay, James E. Tcheng, An-Kwok Ian Wong.

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
