## [Decision Letter · Decision Letter 0]

6 Mar 2024

PDIG-D-23-00357

The High Price of Equity in Pulse Oximetry: A cost evaluation and need for interim solutions

PLOS Digital Health

Dear Dr. Wong,

Thank you for submitting your manuscript to PLOS Digital Health. After careful consideration, we feel that it has merit but does not fully meet PLOS Digital Health's publication criteria as it currently stands. Therefore, we invite you to submit a revised version of the manuscript that addresses the points raised during the review process.

Please submit your revised manuscript within 60 days May 05 2024 11:59PM. If you will need more time than this to complete your revisions, please reply to this message or contact the journal office at digitalhealth@plos.org. Please include the following items when submitting your revised manuscript:

We look forward to receiving your revised manuscript.

Kind regards,

Nan Liu

Academic Editor

PLOS Digital Health

Journal Requirements:

1. Please send a completed 'Competing Interests' statement, including any COIs declared by your co-authors. If you have no competing interests to declare, please state "The authors have declared that no competing interests exist". Otherwise please declare all competing interests beginning with the statement "I have read the journal's policy and the authors of this manuscript have the following competing interests:"

If you did not receive any funding for this study, please simply state: “The authors received no specific funding for this work.

3. Please provide separate figure files in .tif or .eps format only and remove any figures embedded in your manuscript file. Please also ensure that all files are under our size limit of 10MB.

4. We have noticed that you have uploaded Supporting Information files, but you have not included a list of legends. Please add a full list of legends for your Supporting Information files after the references list.

Additional Editor Comments (if provided):

We would like to thank the authors for bringing out an important issue for discussions. While the topic of the study is interesting, there lacks sufficient details and background information to justify its scientific rigor. The authors are suggested to elaborate on various sections of the manuscript to ensure solid methods and convincing results.

Reviewers' comments:

Reviewer's Responses to Questions

**Comments to the Author**

1. Does this manuscript meet PLOS Digital Health’s publication criteria? Is the manuscript technically sound, and do the data support the conclusions? The manuscript must describe methodologically and ethically rigorous research with conclusions that are appropriately drawn based on the data presented.

Reviewer #1: Yes

Reviewer #2: Yes

Reviewer #3: No

2. Has the statistical analysis been performed appropriately and rigorously?

Reviewer #1: Yes

Reviewer #2: Yes

Reviewer #3: No

3. Have the authors made all data underlying the findings in their manuscript fully available (please refer to the Data Availability Statement at the start of the manuscript PDF file)?

Reviewer #1: Yes

Reviewer #2: Yes

Reviewer #3: No

4. Is the manuscript presented in an intelligible fashion and written in standard English?

Reviewer #1: Yes

Reviewer #2: Yes

Reviewer #3: Yes

5. Review Comments to the Author

Reviewer #1: This is a well formulated paper. The findings that “Simply replacing” pulse oximetry hardware to address accuracy disparities may be neither simple, cheap, or timely, and that solutions for addressing pulse oximetry accuracy disparities leveraging current technology may be necessary, are of value to informing current and future service provision.

Reviewer #2: This is an interesting article that extrapolates the expense of replacing pulse oximetry hardware and software across 1 hospital system and the nation itself highlighting the depth of the issue and difficulty in rectifying.

I have a few comments and consideration for revision for the authors:

Intro

-The introduction may benefit from a little more description/explanation of the differences between the different models and what the differences are in terms of considerations when replacing. For example, why are pulse ox modules easier to replace than vital signs monitors

Methods

-it is unclear to me from the methods, exactly what technology would be considered to replace the current pulse ox monitors. My (limited) understanding was that we did not have pulse ox monitors that would overcome the skin color difficulties. But I may be mistaken. When discussing the potential of which equipment was used to calculate costs, it is a little unclear to me what actual equipment/modules/monitors are being considered. If no specific module or monitor is being considered, then I would be interested if the paper more explicitly stated that the authors extrapolated cost of replacement using existing monitor/module equipment models and prices

-It may help the less technologically aware reader, what exactly is meant by the device "integration process." Is this integration with the EMR, computer system, telemetry monitoring/alarm system. All of the above? none of the above?

-I am not sure I understand what is meant by "fleet" is fleet the collection of pulse ox monitors/modules across a hospital system?

-Overall I think the methods was written for someone with technological understanding of what replacement of any hospital system includes and may benefit from revision based on above thoughts

Results:

-Under cost estimates for a single health system, would the authors be able to explain what "specialized care" means - what are the 332 specialized care beds? are they ICU beds?

-The results describe that there are 140 types of pulse ox equipment - in the intro the authors describe 3 types. what is meant by 140 distinct types? distinct companies? Seems surprising to me that one hospital system would have so many types of pulse ox

-missing in results is a discussion of the time it would take for replacement. Understanding that is not the aim of the study, the discussion talks about the time to replacement adn would be curious if the authors had any information on estimating time to replacement

Table 2:

This is a minor point and may not apply to many, but I was a little confused trying to interpret what $20,000 (in millions) refers to. Is there a better way to present the cost in this table?

Reviewer #3: This is an observational cross-evaluation of the cost of a pulse oximetry device and its replacement costs. As a report, I see the value in this type of information. However, it is questionable what this adds to the value of new information. The paper does not reveal the health system the analysis was carried out, and I am unsure if this is questionable as results cannot be replicated or checked elsewhere.

The background provides scant details of why this study is needed. It briefly touches on the racial disparities of pulse oximetry but does not elaborate on the justification of how the results could be used in a policy setting to reduce such issues.

I believe this provides some information as a report to the government or policymakers, but I cannot see the rigor or justification to be presented as a scientific paper.

6. PLOS authors have the option to publish the peer review history of their article (what does this mean?). If published, this will include your full peer review and any attached files.

**Do you want your identity to be public for this peer review?** For information about this choice, including consent withdrawal, please see our Privacy Policy.

Reviewer #1: Yes: Dr Sarah Markham

Reviewer #2: No

Reviewer #3: No

---

## [Editor Report · Decision Letter 1]

18 Jun 2024

The High Price of Equity in Pulse Oximetry: A cost evaluation and need for interim solutions

PDIG-D-23-00357R1

Dear Dr. Wong,

We are pleased to inform you that your manuscript 'The High Price of Equity in Pulse Oximetry: A cost evaluation and need for interim solutions' has been provisionally accepted for publication in PLOS Digital Health.

Best regards,

Po-Chih Kuo, Ph. D.

Section Editor

PLOS Digital Health